# Metabolomics Profiling of Cystic Renal Disease towards Biomarker Discovery

**DOI:** 10.3390/biology10080770

**Published:** 2021-08-13

**Authors:** Dalia Sriwi, Mohamad S. Alabdaljabar, Minnie Jacob, Ahmed H. Mujamammi, Xinyun Gu, Essa M. Sabi, Liang Li, Maged H. Hussein, Majed Dasouki, Anas M. Abdel Rahman

**Affiliations:** 1Department of Biochemistry and Molecular Medicine, College of Medicine, Alfaisal University, Riyadh 11533, Saudi Arabia; dsriwi@alfaisal.edu (D.S.); malabdaljabar@alfaisal.edu (M.S.A.); 2Metabolomics Section, Department of Clinical Genomics, Center for Genomics Medicine, King Faisal Specialist Hospital and Research Centre (KFSHRC), Riyadh 11211, Saudi Arabia; minnie@kfshrc.edu.sa (M.J.); majed.dasouki.md@adventhealth.com (M.D.); 3Clinical Biochemistry Unit, Department of Pathology, College of Medicine, King Saud University, Riyadh 11461, Saudi Arabia; amujamammi@ksu.edu.sa (A.H.M.); esabi@ksu.edu.sa (E.M.S.); 4Department of Chemistry, University of Alberta, Edmonton, AB T6G 2R3, Canada; xinyun@ualberta.ca (X.G.); liang.li@ualberta.ca (L.L.); 5Department of Medicine, King Faisal Specialist Hospital and Research Centre (KFSHRC), Riyadh 11211, Saudi Arabia; hmaged@kfshrc.edu.sa; 6Department of Chemistry, Memorial University of Newfoundland, St. John’s, NL A1B 3X7, Canada

**Keywords:** metabolomics, mass spectrometry, cystic renal disease, dried blood spot

## Abstract

**Simple Summary:**

Cystic renal disease (CRD) is a group of diseases characterized by abnormal sacs, or cysts, in the kidneys. CRD can be detected using certain imaging modalities (i.e., ultrasound). Patients with CRD might be symptoms-free, while others can show symptoms long after cysts development. Although these cysts represent structural changes, we hypothesized that they have an underlying biochemical alteration. If so, this would open the floor for potential biomarker discovery, which would aid in CRD diagnosis and, possibly, treatment. On that basis, this study focuses on identifying biomarkers for CRD. To achieve that, we employed a metabolomics-based approach to identify intermediate molecules inside the cells that are byproducts of biochemical reactions. We used dry blood spots and serum samples of CRD patients and healthy controls to study the differences in their metabolomic profile. Our results suggest that certain metabolites, including uridine diphosphate, cystine-5-diphosphate, and morpholine, are potential biomarkers for CRD. The affected biochemical pathways in CRD include aminoacyl-tRNA biosynthesis, purine, pyrimidine, glutathione, TCA cycle, and some amino acid metabolism. These preliminary results could be the starting point for possible diagnostic and therapeutic approaches for CRD in the future.

**Abstract:**

Cystic renal disease (CRD) comprises a heterogeneous group of genetic and acquired disorders. The cystic lesions are detected through imaging, either incidentally or after symptoms develop, due to an underlying disease process. In this study, we aim to study the metabolomic profiles of CRD patients for potential disease-specific biomarkers using unlabeled and labeled metabolomics using low and high-resolution mass spectrometry (MS), respectively. Dried-blood spot (DBS) and serum samples, collected from CRD patients and healthy controls, were analyzed using the unlabeled and labeled method. The metabolomics profiles for both sets of samples and groups were collected, and their data were processed using the lab’s standard protocol. The univariate analysis showed (FDR *p* < 0.05 and fold change 2) was significant to show a group of potential biomarkers for CRD discovery, including uridine diphosphate, cystine-5-diphosphate, and morpholine. Several pathways were involved in CRD patients based on the metabolic profile, including aminoacyl-tRNA biosynthesis, purine and pyrimidine, glutathione, TCA cycle, and some amino acid metabolism (alanine, aspartate and glutamate, arginine and tryptophan), which have the most impact. In conclusion, early CRD detection and treatment is possible using a metabolomics approach that targets alanine, aspartate, and glutamate pathway metabolites.

## 1. Introduction

A cyst is the most common kidney lesion, appearing in almost 40% of patients undergoing renal imaging [1]. Cystic renal disease (CRD) refers to a heterogeneous group of genetic and acquired disorders characterized by kidneys’ cystic lesions of varying properties. The cystic lesions could be unilateral or bilateral, focal or multifocal, benign or malignant, or acquired or congenital. According to the Bosniak classification, the cystic lesions diversity is mainly based on the cysts’ radiological appearance, which standardizes the characterization and management of CRD. CRD’s diagnosis is mostly radiological and sometimes requires biopsy for pathological confirmation. Ultrasonography can be employed for cysts detection, which is safe and noninvasive. Other modalities include contrast-enhanced computed tomography (CT); with the help of Bosniak classification, cysts can be graded as I, II, IIF, III, or IV based on appearance and chance of malignancy. The grading will also indicate the need for an additional follow-up and the course of management [2].

Although radiology is now considered the gold standard of diagnosis by physicians, there is an increasing demand for more developed high-technological modalities and a shift to personalized medicine. Patient phenotypes and pathological states have opened doors to applying metabolomics in drug discovery processes. They slowly alter modern health care from clinical diagnosis and treatment, based on predictive and preventative health monitoring symptoms, based on the patients’ personalized genomic and metabolic information [3,4]. Metabolomics is a systemic method of identifying and quantifying the complete set of metabolites in a specific sample to achieve a global view of the system’s state [5]. The metabolites measured are typically the intermediate products of various cellular metabolic pathways, profoundly influenced by environmental factors and genomics. Numerous diseases have variable clinical presentations even when they share an underlying etiology. Understanding the complex molecular pathogenesis through molecular profiling could be a critical factor in a standardized disease classification approach. Although genomics helps predict what might happen in the system, it has been demonstrated that metabolomics reveals what is currently happening inside the system [6].

Autosomal dominant polycystic kidney disease (ADPKD), occurring due to PKD1 and PKD2 mutations, have been identified more than three decades ago [7]. However, the exact mechanism underlying ADPKD remains unknown. Significant differences in lipid metabolism expression, between ADPKD patients and healthy controls, were detected in a metabolomics study [7]. Further lipidomics study identified significantly lower diacylglycerol, a byproduct of triglyceride metabolism, levels in mutant kidneys, and cell culture studies demonstrated that Pkd1 deficient renal epithelial cells have a defect in palmitate fatty acid oxidation [8].

Metabolomics analysis of CRD would help us better understand the disease’s biological mechanism and help find diagnostic markers and therapeutic targets. A practical and time-efficient approach is using the Chemical Isotope Labeling (CIL) approach on different sample types, such as urine or blood derivatives, which was well studied previously [9,10]. To the best of our knowledge, no previous studies have assessed the relation between CRDs collectively and metabolomics changes. This study aims at a metabolomics approach to identify specific metabolites that could potentially be used as biomarkers to detect and monitor CRD. This in-depth analysis and quantification, of the metabolites collected from dry blood spot (DBS) and serum samples using mass spectrometry, could offer valuable insight into the cellular pathways affected in cystic renal disease.

## 2. Material and Methods

### 2.1. Elements and Chemicals

The reagents and standard materials used in this study were bought from Sigma Chemicals in St. Louis, Missouri, with a minimum purity of 98%. Pterin (also known as 2-Amino-4-hydroxypteridine) and L-Monapterin were purchased from Schricks Laboratories in Postface, Switzerland. The Isotope-labeled internal standards were purchased from Cambridge Isotope, Inc in Woburn, Massachusetts, and ChromSystems in Grafelfing, Germany. All organic solvents and water utilized in sample and mobile phase preparations were LC-MS grade and secured from Fisher Scientific in Fair Lawn, New Jersey.

The LC-MS grade reagents, including water, acetonitrile (ACN), methanol, and formic acid, were purchased from Fisher Scientific (Ottawa, ON, Canada). Isotope-labeled metabolites were purchased from Sigma Chemicals (St. Louis, MO, USA), Cambridge Isotope, Inc. (Woburn, MA, USA), and Toronto Research Chemicals Inc. (Toronto, ON, Canada), and 13C-dansyl chloride was available from the University of Alberta (http://mcid.chem.ualberta.ca, (accessed on 13 December 2020)).

### 2.2. Study Design, Patient Recruitment, and Sample Collection

Whole blood samples were collected from 7 patients with clinically confirmed CRD during their routine clinical visit. The patients were fasting for at least 12 h. A 100 µL of the whole blood was spotted on Guthrie cards (Perkin Elmer 226) upon arrival at the lab, and the cards were stored at −20 °C. The rest of the blood sample was centrifuged to obtain serum fluid, and the aliquots were stored at −80 °C for metabolomics analysis. In summary, for DBS analysis, DBS (n = 7) were collected from clinically confirmed CRD patients and (n = 7) healthy adult volunteers, whereas for serum analysis (n = 7) were collected from clinically confirmed CRD patients and (n = 33) healthy control (Ctrl) serum samples were collected different form studies. The final protocol was revised and approved by the Research Ethics committee of King Faisal Specialist Hospital and Research Center (KFSHRC) Project # 2160027. Patients were chosen at random by the recruiting clinician, and whole blood samples were collected. Any patient unable or unwilling to provide informed consent was excluded from this study.

### 2.3. Label-Free Metabolomics Profiling

Metabolomics analysis on LC-MSMS consisting of a panel of 220 clinically relevant metabolites (amino acids, sugars, organic acids, bile acids, acylcarnitines, neurotransmitters, polyamines, and steroids) was completed as explained previously [11].

Patients, healthy Ctrl, and pooled blood DBS samples were punched (5 discs, 1/8 inch each) into a 96 well plate (in duplicates). The metabolites were extracted by adding 150 μL extraction solvent (40% methanol, 40% acetonitrile, 20% H_2_O) to each well, as described previously (Figure 1). The supernatant was transferred to another plate and evaporated. The extract was then reconstituted by adding 100 μL of the initial condition mobile phase and then stored for LC-MS/MS analysis at 4 °C. 

The extracted metabolites were divided via reversed-phase chromatography. Using Acquity UPLC C18, 1.7 µm, 2.1 × 100 mm^2^ column, a negative and positive ionization mode analysis was performed. The mobile phase consisted of (A) 0.1% acetic acid and (B) 50% acetonitrile (ACN) and 50% Methanol (MeOH) for the positive mode, whereas in negative mode, the mobile phase was composed of (A) 0.1% tributylamine (TBA), 0.03% acetic acid, 10% MeOH and (B) 100% ACN, with a run time of 15 min at a flow rate of 0.3 mL/min.

LC-MSMS analysis was performed as detailed elsewhere, [11] where the source and desolvation temperatures were set at 150 °C and 500 °C, respectively. The desolvation gas was set at 1000 L/h in both the polarity modes. The cone voltage fluctuated from 18 to 170 V, and the collision energy ranged from 7 to 65 eV. 

LC-MSMS data was processed using Target Lynx WATERS-S4RWRX6, MassLynx V4. 1. Ink (Waters Corporation, Milford, MA, USA). The software achieved data analysis and peak integration. Our main analytical indication was the area measured under the peak. 

### 2.4. CIL LC-MS Metabolomics Profiling on Serum for CRD Patients

The Chemical isotope labeling liquid chromatography-mass spectrometry (CIL LC-MS) approach labels phenol amine submetabolome by 12C dansyl-chloride (DnsCl). In the other hand, a pooled sample was labeled by 13C DnsCl as a reference for the 12C-labeled samples as described earlier (Figure 1) [12]. The quality control (QC) samples were injected once every 15 runs, where Peak pairs with ratio values having > ±25% RSD in the QC samples were filtered out. The metabolic extracts were analyzed using a reversed-phase chromatography, C18 column (2.1 mm × 10 cm, 1.8 μm particle size, 95 Å pore size (Agilent Inc., Santa Clara, CA, USA) on Thermo Fisher Scientific Dionex Ultimate 3000 UHPLC System (Sunnyvale, CA, USA) linked to a Bruker Maxis II quadrupole, time-of-flight (Q-TOF) mass spectrometer (Bruker, Billerica, UK). The mobile phase A constituted of 0.1% (*v*/*v*) formic acid in 5% (*v*/*v*) ACN, while solvent B was 0.1% (*v*/*v*) formic acid in acetonitrile. 

The CIL LC-MS serum metabolomics profiling was processed by Bruker Daltonics Data Analysis 4.3 Software. Peak pairs were extracted from CSV files by IsoMS, removing the unwanted pairs. Data were aligned based on the peak’s accurate mass and retention time, and zero-fill software was used to fill up the missing values in the aligned file [13,14]. The significantly differentially expressed metabolites were identified by performing a univariate analysis (volcano plot), where we used a criterion of the fold change of greater than 2. PLS-DA plots were performed using Iso MS Pro. (NovaMT Inc., Edmonton, AB, Canada). The metabolites were positively identified by searching against DnsID Library (www.mycompoundid.org, (accessed on 13 December 2020)) using retention time and accurate mass [14]. Putative identification was performed by searching accurate mass against the MyCompoundID library (www.mycompoundid.org, (accessed on 13 December 2020) [15].

### 2.5. Statistical Analysis

The raw data from the metabolomics experiments were statistically analyzed using MetaboAnalyst software version 4.0 (McGill University, Montreal, QC, Canada) [16]. Analytes with missing data points for more than 80% were filtered out. In contrast, some rows with missing values were replaced with small values (half of the minimum positive values in the original data) assumed to be above the detection limit.

The data were normalized to the equivalent internal standard’s area under the peak and the total sample median for normal distribution evaluation. The samples’ differences were adjusted by data log transformation. Pareto scaling approaches to ensure a comparative review for individual features. The significance of metabolomics data was evaluated at an FDR-corrected *p*-value < 0.05, where the values were reported as mean ± SEM. The chemometric analysis used orthogonal partial least-squares projection to latent structure discriminant analysis (OPLS-DA) and supervised multiple regression analysis to identify the discrimination between different datasets [17]. The potential biomarkers and features of the study groups were used for pathway analysis. The Receiver Operating Characteristic (ROC) curves were constructed using the PLS-DA method in the MetaboAnalyst software version 4.0 for global analysis. 

## 3. Results

### 3.1. Demographics, Clinical and Molecular Features in CRD Patients

The clinical and laboratory characteristics of the study cohorts are represented in Table 1. The current study included DBS samples collected from CRD patients (n = 7) and healthy controls (Ctrl) (n = 7). Another serum sample was collected from the same CRD patients (n = 7) and a bigger group of healthy controls (n = 33). As shown in Table 1, the patients’ ages ranged between 8–78 years, with a mean of 40.4 ± 24.8 years (SD). Each patient has different CRD phenotypes; however, all of them showed evidence of renal cysts. The patients’ eGFR ranges between 9–92 mL/min, with only one (CRD-7) with a normal eGFR and two patients (CRD-2 and 3) with an eGFR falling in the kidney failure range. The seven patients showed different comorbidities, which might be related to the primary renal pathology (e.g., ADPKD and hypertension in CRD-3). Noteworthy, almost half of CRD patients have hypertension.

### 3.2. Metabolomics Pattern in DBS of CRD Patients

The raw data of the average replicates were normalized, transformed, and scaled by a median, log, and Pareto, respectively, to ensure that all visualized data are under Gaussian distribution. Appendix A shows the normalization of metabolites in CRD versus Ctrl groups in DBS to remove systemic variations. The study comorbidities’ effects were evaluated using the multi-binary analyses between the CRD sub-groups, i.e., CRD+HTN vs. CRD-HTN, and we could not find any significant metabolic associated with these conditions.

Figure 2A represents dysregulated detected features. Among them, 17 metabolites were up-regulated, and 15 were down-regulated based on all the detected features that were differentially expressed. The cut-off values for the p-value and fold change were 0.05 and 2, respectively. In Figure 2B, Orthogonal partial least squares discriminant analysis (OPLS-DA) demonstrates the variability between CRD and Control groups. The analysis was performed in duplicates for both study groups. The score plot depicts the difference between both groups evident by the apparent separation, with a calculated R2 = 0.998 and Q2 = 0.726. The group separation represents the variability in metabolomics expression level between groups, which might be explained by the presence and absence of renal cysts in CRD and Control groups.

Based on OPLS-DA data, a Selected Frequency % score was generated to identify individual metabolites’ contribution levels. Possibly, this can pave the ground for biomarker discovery for CRD patients. The selected frequency plot (Figure 2C) shows pyrimidines, such as uridine diphosphate (UDP), cytidine diphosphate (CDP), and guanine monophosphate (GMP) to be up-regulated in CRD patients in comparison to the Control group. In pathway analysis (Figure 2D), amino acids (alanine, aspartate, and glutamate), purine and pyrimidine metabolism, glutathione metabolism, and TCA cycle pathways were significantly altered. These changes are represented by the circles (pathway impact) and size (statistical significance, p-value) on the figure.

The heat map (Appendix A) represents the relative concentration of metabolites in CRD compared to Control groups, expressed in different red and green colors intensities, respectively. Appendix A summarize statistically significant pathways and metabolites between CRD and Ctrl groups.

### 3.3. Metabolomics Pattern in Serum of CRD Patients

Similarly, univariate analyses showed that differentially expressed metabolites were visualized on the volcano plot (Figure 3A). One hundred six metabolites were up-regulated and 70 downregulated in CRD when compared to the Control group. Unfortunately, among the 7 serum samples, one sample (CRD7) was unavailable for analysis due to technical issues related to the pre-acquiring sample collection and processing. The cut-off *p*-value has a corresponding *q*-value of less than 0.05 and a fold change cut-off value of 2. Figure 3B, which represents OPLS-DA, shows the discrimination between both groups, with an estimated R2 of 0.994 and Q2 of 0.948, which indicate the differences in metabolites expression among CRD and Control groups. The Selected frequency score plot in Figure 3C demonstrates that morpholine, leucyl glutamine, and isoleucyl aspartate were up-regulated in CRD compared to the Control group. Lastly, pathway analysis (Figure 3D) depicts the most significantly affected pathways, including tRNA biosynthesis and amino acid metabolism pathways. It is worth mentioning that the tRNA biosynthesis pathway was among the commonly impacted pathways in metabolomics profiling of DBS and serum (Figure 1D and Figure 2D). The heat map in Appendix A shows altered metabolites in DBS and serum, respectively, among the CRD versus Control groups, whereas Appendix A summarize statistically significant changes in pathways and metabolites. The heat map in Appendix A shows altered metabolites in DBS and serum, respectively, among the CRD versus Control groups, whereas Appendix A summarize statistically significant changes in pathways and metabolites.

### 3.4. Biomarker Evaluation

Receiver operating characteristics (ROC) exploring curves were generated from the binary comparisons between CRD patients and Control groups. Multivariate exploratory ROC analysis was generated using PLS-DA as a classification and feature ranking method. The combination of the top-ranked metabolites in ROC curves shows AUCs ranging from 0.995–0.99 (Figure 4A) (Appendix A). The significant features of the expressed metabolites in the CRD patients and Control groups are represented in the selected frequency plots (Figure 4B). Ornithine (Figure 4C) and 2,3-Pyridinedicarboxyic acid (Figure 4D) were down-regulated in CRD patients compared to the Control group.

Similarly, multivariate exploratory ROC analysis in serum showed AUCs ranging from 0.995–1 (Figure 5A) (Appendix A). The significant features of the positively identified metabolites are represented in a selected frequency plot (Figure 5B): 2-Amino-4-chloro-4-pentenoic acid (Figure 5C) and 2-[(2-Aminoethylcarbamoyl) methyl]-2-hydroxybutanedioic acid (Figure 5D) to be up-regulated in the CRD patients when compared to the Control group. Using accurate mass and retention time matches to metabolite entries in the MyCompoundID standard library, 2-Amino-4-chloro-4-pentenoic acid was identified, including most of the KEGG library compounds. The KEGG number for this compound is C04075. Similarly, we identified 2-[(2-Aminoethylcarbamoyl) methyl]-2-hydroxybutanedioic acid using mass and retention time matches, and the KEGG number is C21559.

## 4. Discussion

This study focused on identifying CRD specific metabolomic pathways that could be utilized in disease detection and follow-up. Diverse panels of metabolites were used to cover multiple cellular pathways in both DBS and serum, such as glycolysis, tricyclic acid, and pentose phosphate pathways.

All cystic renal diseases share the same central pathology of a cyst, a fluid-filled sac [18]. However, many other factors determine the morbidity and mortality of each CRD, including, but not limited to, age, sex, family history of renal malignancy, genotype, and presence of other comorbidities. For instance, simple cysts appear on ultrasound as a solitary anechoic mass and concern malignant transformation if they grow in size, bleeds, or get infected [19]. On the other hand, ADPKD is usually clinically apparent due to the significant increase in both kidneys, which can be associated with hypertension features, and rarely, intra-abdominal bleeding [12].

One potential and promising technique to study disease for diagnostic and therapeutic purposes is metabolomics [4]. Metabolomics is an advanced science field that measures small molecules’ overall expression (<1500 Da). Similarly, genome, transcriptome, and proteome focus on other bio-substrate synthesis and metabolism. The study of metabolites can aid in disease detection and, theoretically, treatment. The dysregulation of specific metabolites can signal an altered metabolic pathway, which might have great potential as a therapeutic target. Metabolomics application in real life has been extensively studied for biomarker detection, and early diagnosis, of multiple diseases [5,20,21].

This study aims to introduce potential metabolic biomarkers for CRD detection and prognosis, which is the most common inherited kidney disease and accounts for 4.5% of all end-stage renal disease cases [5]. Many studies on renal diseases have used a metabolomics approach to understand disease pathogenesis [20,22,23,24,25]. Among different studies that looked into kidney diseases and metabolomics, few have focused on PRD in rats, [26,27] mice [28], and humans [29,30].

A metabolomics study identified a pathogenic metabolic pathway using ADPKD as a therapeutic target [31]. However, the glucose metabolism pathway results in enhanced glycolysis in Pkd1 defective cells. There is no balance between proliferation and apoptosis in ADPKD tissues, and glucose deprivation restored the proliferation index by lowering the proliferation and increasing apoptotic rates in Pkd1 mutant cells. The increased ATP content in mutated cells was attributed to aerobic glycolysis, and most enzymes involved in glycolysis were up-regulated, while enzymes involved in gluconeogenesis were downregulated [31]. In our study, nucleotide phosphate molecules (UDP, CDP, GMP), certain amino acids, and morpholine were found to be up-regulated in CRD patients. This metabolite panel does not contradict the previous study’s findings but suggests that other pathways might be affected. It’s noteworthy that our study’s findings are based on patients with different CRD phenotypes rather than ADPKD alone, which might also explain the difference in the pathway and metabolites analysis.

Metabolomics profiling in serum identified an alteration in the aminoacyl-tRNA biosynthesis in common. The metabolomics analysis has also detected an alteration in the amino acids’ metabolic pathways: alanine, aspartate, and glutamate. The CIL metabolomics analysis has further identified the downregulation of 2-Amino-3-carboxymuconate and Glutamine and Morpholine’s upregulation in CRD patients. Comparing two datasets is difficult because the platforms cover different metabolites expressed differently in different sample types. However, Aminoacyl-tRNAs are essential substrates for protein translation attached to an amino acid by an aminoacyl-tRNA synthetase. Aminoacyl-tRNAs are crucial substrates for protein translation that bind to an amino acid by an aminoacyl-tRNA synthetase. The amino acid identity inserted into the protein molecule is determined by the mRNA codon paired with that specific tRNA molecule [11]. The amino acid identity, inserted into the protein molecule, is determined by the mRNA codon paired with that particular tRNA molecule [32]. Amino acids have essential roles in metabolic pathways, as substrates and regulators, and have been previously investigated [33]. Our pathway analysis identified significant differences between the CRD and the Control groups in the biosynthesis of alanine, aspartate, and glutamate, 2-Amino-3-carboxymuconate glutamine, and morpholine.

Although cysts are structural pathologies, they have underlying pathophysiology, expected to develop other cells’ alterations. For example, PKD was widely studied before using metabolomics. Abbiss et al. (2019) have summarized the known connected metabolic dysregulations in PKD to include allantoin, 2-hydroxyglutarate, 2-oxoglutaric acid, aconitic acid, ADMA, carnitine, citrate, creatinine, hippuric acid, malic acid, myo-inositol, trimethylamine oxide, uric acid, 3-indoxyl sulfate, 3-methylhistidine, acetylcarnitine, citrulline, fumaric acid, glutamic acid, glutamine, glycine, hypoxanthine, N, N-dimethylglycine, pantothenic acid, pipecolate, and trigonelline [34]. This spectrum of metabolic dysregulations shows the wide variety of metabolites impacted in one cystic disease, PKD. Similarly, in this study, a group of amino acids was somehow linked to cyst development, as reported recently in nonhuman-based studies [26,27,30,35].

Elevated levels of amino acids (alanine, aspartate, and glutamate) have been found in patients with type 2 diabetes [36,37]. In contrast, leucine has a positive association with clear cell renal cell carcinoma [38]. There are significant biochemical similarities between CRD and malignant processes, both associated with aberrant cell proliferation, which justifies using antiproliferative drugs to treat certain cystic kidney diseases [39]. Previous metabolomic reports have detected a decreased arginosuccinate synthase activity in the proximal tubules of ADPKD compared to renal cell cancer patients [35]. Renal cell carcinoma and PKD exhibit cyclin-dependent kinase inhibitors (CKIs) abnormalities are at the core of their pathology. CKIs, such as p21, have a significant role in activating apoptosis in dysfunctional cells. One study specifically examined p21, in transgenic rat models with ADPKD, and found it decreased to levels similar to malignant cells [40]. Some metabolic pathways between CRD (ARPKD, specifically) and cancer were also suggested by Hwang et al. [30]. The above should not come as a surprise, knowing that the antiproliferative mTOR inhibitors have long been considered a potential therapeutic intervention [41].

Morpholine is a building block in preparing antibiotics like linezolid and the anticancer agent gefitinib (Iressa). Morpholine, also called acu-dioxomorpholine, is positively linked to P-glycoprotein inhibitors in multi-drug resistant cancers [42]. Another study identified N-[2-hydroxy-1-(4-morpholinylmethyl)-2-phenylethyl]-decanamide monohydrochloride (DL-PDMP) to inhibit the proliferation of aneuploid colorectal cancer cells selectively [43]. It appears that there is no clear link between the upregulation of morpholine and CRD pathology in previous literature. Morpholine is also an industrial chemical with potential toxicity to the kidney and liver and may accumulate when renal function is impaired, as in several of our patients with CRD. We used accurate mass and retention time matches to identify morpholine in the MyCompoundID library (HMDB0031581 and KEGG C14452). Morpholine is a component of edible coatings for fruit and vegetables and its usage as a water additive. Using our sensitive CIL LC-MS method, it is not surprising to detect this compound in blood.

This study has some limitations, where the sample size is small and heterogeneous in etiology and the degree of renal impairment. For instance, based on this cohort, it will be quite difficult to correlate the eGFR effect on the unique metabolic expression for the CRD. Despite these limitations, both metabolomics platforms, and the two sample types, have shown clusters of differentially regulated metabolites for CRD compared to the healthy group. Based on the application and clinical validation study, the DBS is an ideal sample type that needs to be further evaluated in a prototype model. The implications are that the cysts are not just a metabolically neutral imaging phenomenon but are associated with distinct metabolomic alterations. The exact relationship between these alterations and the process of cystogenesis remains unclear. Potential confounders, which might have contributed to our findings, need to be considered in future work, as reported recently for more reliable markers. In terms of methodology, both metabolomics approaches are targeted to the limited number of metabolites.

Moreover, since this is the first report showing the holistic metabolomics picture in CRD’s, linking it to other previously reported findings has not been possible. The currently available studies mainly focus on ADPKD, which is unrepresentative in our cohort (n = 1). These factors have to be considered in a future follow-up study for better CRD biomarker identification.

## 5. Conclusions

In conclusion, cysts can affect kidneys, which results in CRD, a common renal pathology. CRD encompasses a broad spectrum of renal diseases, which are usually diagnosed by imaging, mainly ultrasound. This study used metabolomics approaches, in DBS and serum, to look for possible biomarkers to aid in CRD diagnosis. In summary, we found that aminoacyl-tRNA biosynthesis, branched amino acids, and tryptophan metabolism are among the positively impacted pathways in the CRD group. In addition, nucleic acids, including UDP, CDP, and GMP, were up-regulated in the CRD group in DBS metabolomics analysis. Our findings suggest a promising group of biomarkers for CRD diagnosis, but further studies are needed to confirm these biomarkers. They test their ability to serve as a diagnostic tool for CRD.

## Figures and Tables

**Figure 1 biology-10-00770-f001:**
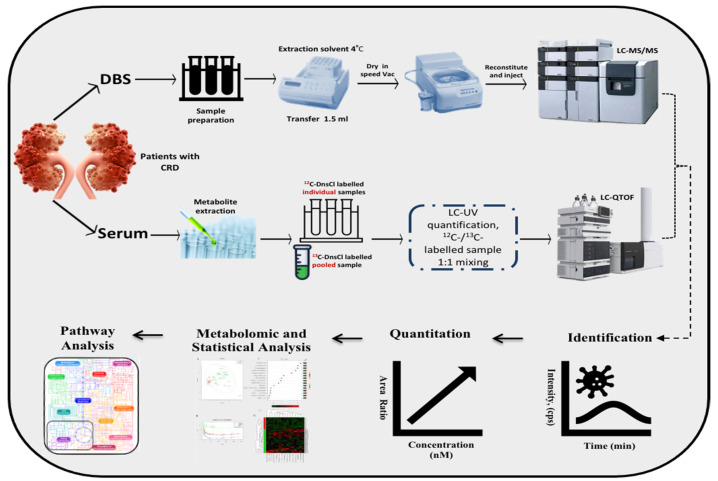
Metabolomics workflow.

**Figure 2 biology-10-00770-f002:**
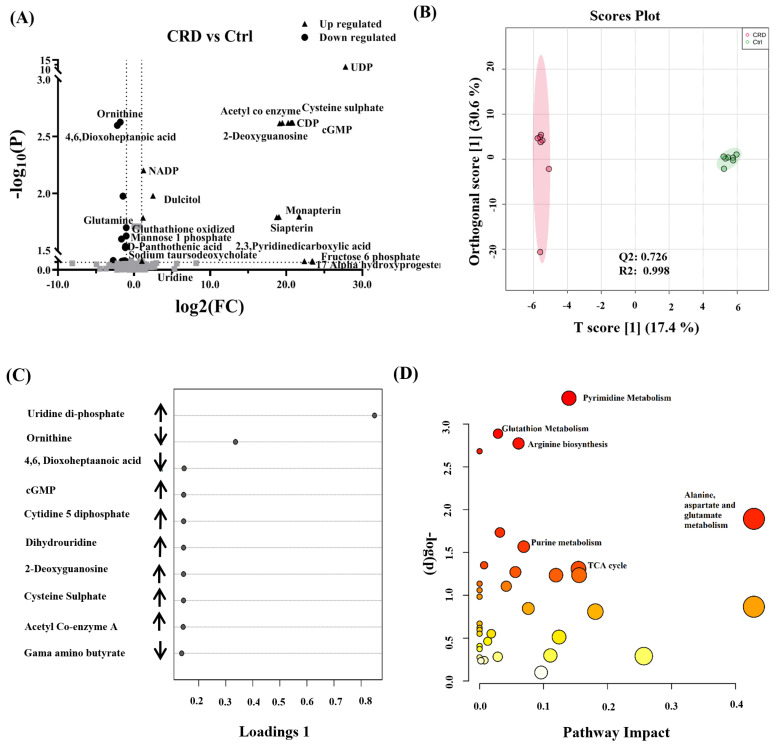
Binary comparison and pathway analysis in DBS of patients with cystic renal diseases (CRD). (**A**) Dysregulated metabolites between CRD vs. Ctrl with fold change cut-off 2 (up-regulated (n = 17) down-regulated (n = 15)) based on all detected features (**B**) Orthogonal partial least square discriminant analysis (OPLS-DA) score plot shows apparent clustering and separation, due to the biological differences between patients with CRD (n = 7) and healthy Ctrl (n = 7) (R2 = 0.998 and Q2 = 0.726) that represents the level of global metabolic differential expression (sample run in duplicates). (**C**) Loading plot shows the regulation of metabolic expression of metabolites in CRD compared to the Ctrl group. (**D**) Pathway analysis shows significantly altered amino acid metabolism and purine and pyrimidine, glutathione, and TCA cycle pathways. (color-coded and the circle’s size reflect the combination between the *p*-value and the pathway impact, respectively.).

**Figure 3 biology-10-00770-f003:**
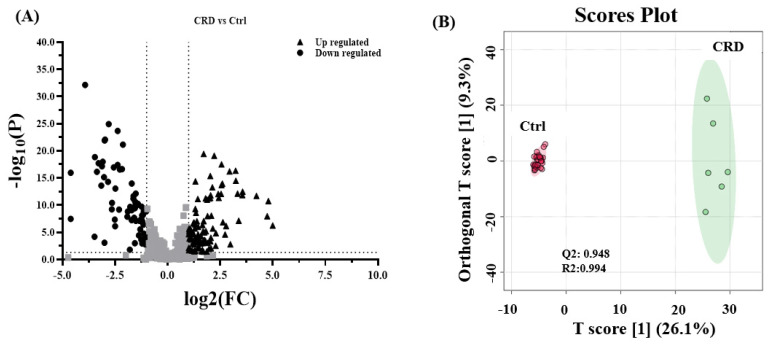
Binary comparison and pathway analysis and in the serum of patients with cystic renal diseases. (**A**) A Volcano plot analysis with fold change > 2 (up-regulated (n = 106), down-regulated (n = 70)) (**B**) Orthogonal partial least square discriminant analysis (OPLS-DA) score plot shows clear separation due to the biological differences between patients with CRD (n = 6) and Ctrl groups (n = 33) (R2 = 0.994 and Q2 = 0.948), that represents the level of global metabolic differential expression. (**C**) The loading plot shows the regulation of metabolic expression of metabolites in CRD compared to the Ctrl group. (**D**) Pathway analysis shows significantly altered pathways of Aminoacyl-tRNA biosynthesis and some amino acid metabolism pathways. (color-coded and the circle’s size reflect the combination between the *p*-value and the pathway impact, respectively).

**Figure 4 biology-10-00770-f004:**
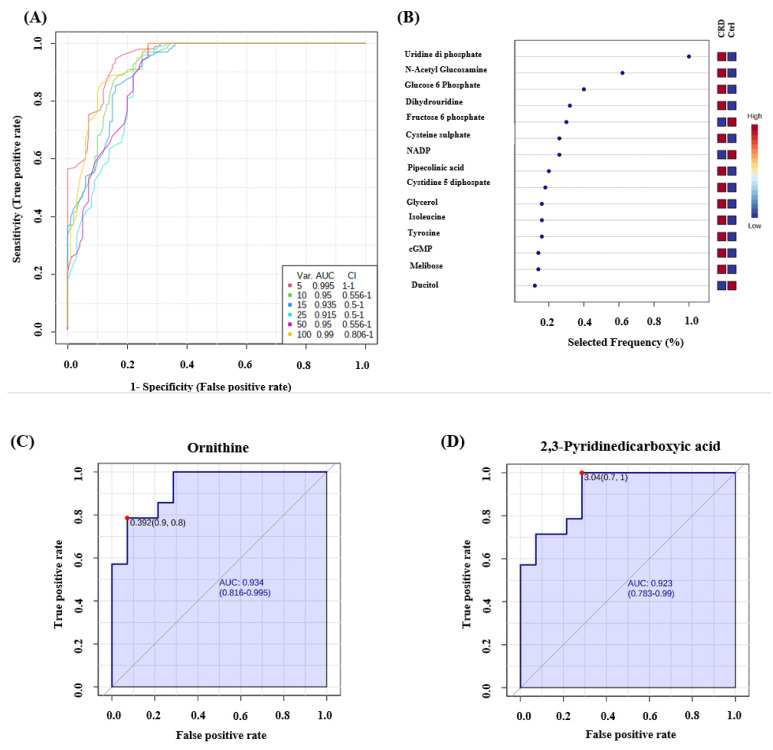
Receiver operating characteristics (ROC) curve and loading plots for significant metabolites in DBS of patients with cystic renal diseases. (**A**) ROC was generated by PLS-DA model showing area under the curve (AUC) for the top five variants = 0.995. (**B**) Frequency percentage plot of the dysregulated metabolites in CRD patients when compared to the controls. (**C**) Ornithine (AUC-0.934) and (**D**) 2,3-Pyridinedicarboxyic acid (AUC-0.923) were down regulated in CRD patients Data were normalized, transformed, and scaled by median, log, and Pareto scaling to make sure all the data are under Gaussian distribution.

**Figure 5 biology-10-00770-f005:**
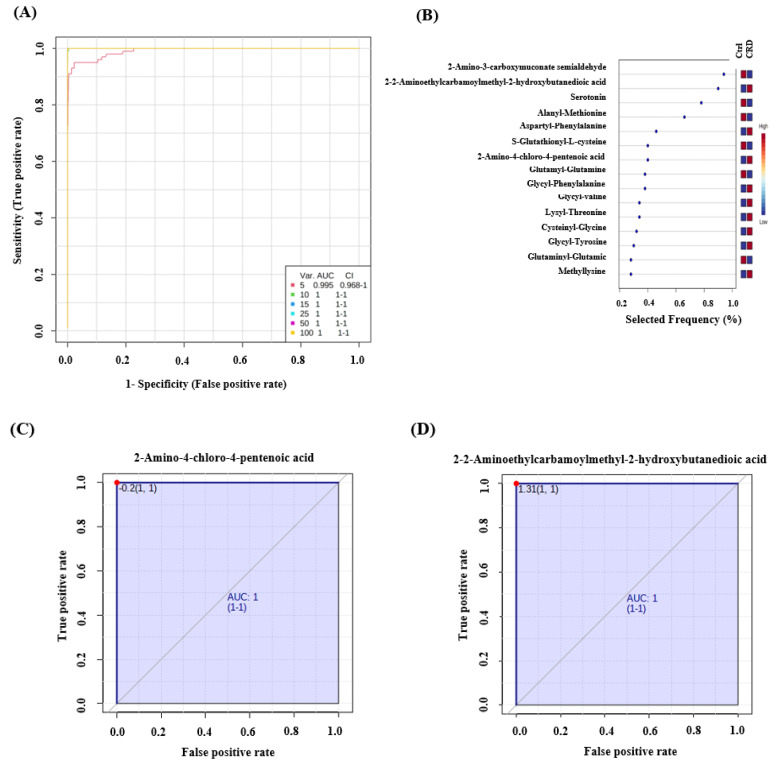
Receiver operating characteristics (ROC) curve and loading plots for significant metabolites in serum patients with cystic renal diseases. (**A**) ROC was generated by the PLS-DA model showing area under the curve (AUC) = 0.995–1 (**B**) Loading plots showed frequently expressed metabolites in CRD patients compared to the Ctrl group. (**C**) 2-Amino-4-chloro-4-pentenoic acid and (**D**) 2-[(2-Aminoethylcarbamoyl) methyl]-2-hydroxybutanedioic acid are up-regulated in CRD patients (AUC-1).

**Table 1 biology-10-00770-t001:** Summary of relevant clinical characteristics of patients in CRD group CRD: Cystic Renal Disease; CKD: Chronic Kidney Disease; DM: diabetes mellitus; eGFR: estimated glomerular filtration rate; ESRD: End-Stage Renal Disease; HCV: Hepatitis C Virus; HTN: Hypertension.

Patient	Age (Yrs)	CRD Phenotype/Renal Disease	Other Comorbidities	eGFR (mL/min)
CRD-1	8	Renal hypodysplasia/Facial dysmorphism	N/A	46
CRD-2	20	Autosomal recessive polycystic kidney disease	ERSD on HD, autoimmune thrombocytopenia, congenital hepatic fibrosis	10
CRD-3	45	Autosomal dominant polycystic kidney disease/Failed kidney transplant	DM, HTN, dyslipidemia, chronic HCV	9
CRD-4	22	Cystic hypokalemic nephropathy/Apparent Mineralocorticoid excess	Congenital adrenal hyperplasia	74
CRD-5	78	Bilateral cortical simple cysts/Diabetic kidney disease	DM, HTN, HNF1B mutation	46
CRD-6	51	Bilateral cortical simple cysts/CKD	Valvular heart disease	39
CRD-7	59	Bilateral cortical simple renal cysts/Focal segmental glomerulosclerosis, CKD	DM, HTN, HNF-B mutation, proteinuria	92

## Data Availability

The data that support the findings of this study are available from the corresponding author upon reasonable request.

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
