# Peer review of "Metabolomics Profiling of Cystic Renal Disease towards Biomarker Discovery"

_biology, 2021, doi:10.3390/biology10080770_

Round 1
Reviewer 1 Report
In their work authors explore the metabolomic profiles of two different samples (DBS and serum) from patients with cystic renal disease (7) and controls (7 to 33) to verify if there are some biomarkers able to identify the former group non-invasively. Although previous examples demonstrated the dysregulation of some metabolic pathways in ADPKD patients, less is known about the modifications that can be encountered in all the other causes of cystic renal disease, so the proposed study could be of some interest and eventually impact in the future. However, some remarks about the design of the study exists:
- the "cases" cohort is quite small and less is known about the genetic landscape of the CRD of these cases (2 have HNF-B mutation and one should have ADPKD, although this last is not specified in Table 1), and could be of interest to link the metabolic "signature" found with the specific genetic alteration
- all the cases have common morbidities (e.g. all have hypertension) and less is known about the clinical characteristics of controls.. if none of the controls have hypertension, in example, we could not exclude that the metabolomic differences encountered between the two groups can be referred to this "confounding" factor
- many of the cases have a decreased glomerular filtration rate (eGFR), which means that their kidneys are unable (or less efficient) in filtering the blood, which inevitably can cause alterations in the blood composition, hence metabolomic signature.. this could be at least mentioned as a factor that can affect the analysis and as a possible source of inter-individual variability
- after reading the paper, it is not clear which sample should be the "promising" one for future testing, if the DBS or the serum, since they led substantial different results during the metabolomic analysis
After these observations, here are my minor remarks on the manuscript:
- Introduction, page 2, line 72, authors should use the complete form for controls, not ctrls
- Figure 1A, the labels of the spots are too small to be read and partially overlap, authors should address this
- Results, page 3, line 104-109, these are more methods than results, consider moving this part in the appropriate section
- Results, page 5, line 140, authors state that one serum sample was unavailable for the analysis, but should explain why as well
- Discussion, page 10, line 249 and 260-263, sentences or words are cancelled
Author Response
Please see our response in the attached file

Reviewer 2 Report
In the current manuscript, Sriwi et al., aimed to study the metabolomic profiles of CRD patients in light of finding a potential disease-specific biomarkers using mass spectrometry methods. Dried-blood spot (DBS) and serum samples were collected from CRD patients and healthy controls and analyzed. In short, the aim of the study is clear, however, the reviewer has few concern about the significant outcome of the study.
Major points,
- The author has used only 7 CRD samples for analysis, with heterogeneous characteristics, so, how does the author consider the interpretations made from the study is reliable?, as metobolomic profiles could be easily altered by different comorbidities.
- The author has used two types of sampling for their metabolic study, what is the rationale of using two different samples?
- Fig1 & 2, why the pathway analysis in DBS and serum sample from the same patients are different? E.g the orthogonal score vs T score is different between control vs CRD samples. Why the author did performed different pathway profiling in DBS and serum samples of the same patients. Do the author determined pathway analysis in all patients?
- 4 Biomarker evaluation/Fig 3 & 4, how does the author selected the metabolites for consideration? Why some of the top metabolites with high AUC are not considered for the loading plots or ROC curves.
- The text “Ctrl” should be elaborated throughout the manuscript.
Author Response
Please find the authors' response in the attached file

Round 2
Reviewer 1 Report
Authors addressed the majority of the reviewers observations
Reviewer 2 Report
The revised manuscript improved significantly and the authors addressed all reviewers concern satisfactorily. The manuscript can now be considered for publication.